# Dimensionality Reduction with Subspace Structure Preservation

**Devansh Arpit**
Department of Computer Science
SUNY Buffalo
Buffalo, NY 14260
devansha@buffalo.edu

**Ifeoma Nwogu**
Department of Computer Science
SUNY Buffalo
Buffalo, NY 14260
inwogu@buffalo.edu

**Venu Govindaraju**
Department of Computer Science
SUNY Buffalo
Buffalo, NY 14260
govind@buffalo.edu

## Abstract

Modeling data as being sampled from a union of independent subspaces has been widely applied to a number of real world applications. However, dimensionality reduction approaches that theoretically preserve this independence assumption have not been well studied. Our key contribution is to show that $2K$ projection vectors are sufficient for the independence preservation of any $K$ class data sampled from a union of independent subspaces. It is this non-trivial observation that we use for designing our dimensionality reduction technique. In this paper, we propose a novel dimensionality reduction algorithm that theoretically preserves this structure for a given dataset. We support our theoretical analysis with empirical results on both synthetic and real world data achieving *state-of-the-art* results compared to popular dimensionality reduction techniques.

## 1 Introduction

A number of real world applications model data as being sampled from a union of independent subspaces. These applications include image representation and compression [7], systems theory [13], image segmentation [16], motion segmentation [14], face clustering [8, 6] and texture segmentation [9], to name a few. Dimensionality reduction is generally used prior to applying these methods because most of these algorithms optimize expensive loss functions like nuclear norm, $\ell^1$ regularization, e.t.c. Most of these applications simply apply off-the-shelf dimensionality reduction techniques or resize images (in case of image data) as a pre-processing step.

The union of independent subspace model can be thought of as a generalization of the traditional approach of representing a given set of data points using a single low dimensional subspace (e.g. Principal Component Analysis). For the application of algorithms that model data at hand with this independence assumption, the subspace structure of the data needs to be preserved after dimensionality reduction. Although a number of existing dimensionality reduction techniques [11, 4, 1, 5] try to preserve the spacial geometry of any given data, no prior work has tried to explicitly preserve the independence between subspaces to the best of our knowledge.

In this paper, we propose a novel dimensionality reduction technique that preserves independence between multiple subspaces. In order to achieve this, we first show that for any two disjoint subspaces with arbitrary dimensionality, there exists a two dimensional subspace such that both the

subspaces collapse to form two lines. We then extend this non-trivial idea to multi-class case and show that $2K$ projection vectors are sufficient for preserving the subspace structure of a $K$ class dataset. Further, we design an efficient algorithm that finds the projection vectors with the aforementioned properties while being able to handle corrupted data at the same time.

## 2   Preliminaries

Let $S_1, S_2 \ldots S_K$ be $K$ subspaces in $\mathbb{R}^n$. We say that these $K$ subspaces are independent if there does not exist any non-zero vector in $S_i$ which is a linear combination of vectors in the other $K - 1$ subspaces. Let the columns of the matrix $B_i \in \mathbb{R}^{n \times d}$ denote the support of the $i^{th}$ subspace of $d$ dimensions. Then any vector in this subspace can be represented as $x = B_i w \;\; \forall w \in \mathbb{R}^d$. Now we define the notion of margin between two subspaces.

**Definition 1** *(Subspace Margin) Subspaces $S_i$ and $S_j$ are separated by margin $\gamma_{ij}$ if*

$$\gamma_{ij} = \max_{u \in S_i, v \in S_j} \frac{\langle u, v \rangle}{\|u\|_2 \|v\|_2} \tag{1}$$

Thus margin between any two subspaces is defined as the maximum dot product between two unit vectors $(u, v)$, one from either subspace. Such a vector pair $(u, v)$ is known as the principal vector pair between the two subspaces while the angle between these vectors is called the principal angle.

With these definitions of independent subspaces and margin, assume that we are given a dataset which has been sampled from a union of independent linear subspaces. Specifically, each class in this dataset lies along one such independent subspace. Then our goal is to reduce the dimensionality of this dataset such that after projection, each class continues to lie along a linear subspace and that each such subspace is independent of all others. Formally, let $X = [X_1, X_2 \ldots, X_K]$ be a $K$ class dataset in $\mathbb{R}^n$ such that vectors from class $i$ ($x \in X_i$) lie along subspace $S_i$. Then our goal is to find a projection matrix ($P \in \mathbb{R}^{n \times m}$) such that the projected data vectors $\bar{X}_i := \{P^T x : x \in X_i\}$ ($i \in \{1 \ldots K\}$) are such that data vectors $\bar{X}_i$ belong to a linear subspace ($\bar{S}_i$ in $\mathbb{R}^m$). Further, each subspace $\bar{S}_i$ ($i \in \{1 \ldots K\}$) is independent of all others.

## 3   Proposed Approach

In this section, we propose a novel subspace learning approach applicable to labeled datasets that theoretically guarantees independent subspace structure preservation. The number of projection vectors required by our approach is not only independent of the size of the dataset but is also fixed, depending only on the number of classes. Specifically, we show that for any $K$ class labeled dataset with independent subspace structure, only $2K$ projection vectors are required for structure preservation.

The entire idea of being able to find a fixed number of projection vectors for the structure preservation of a $K$ class dataset is motivated by theorem 2. This theorem states a useful property of any pair of disjoint subspaces.

**Theorem 2** *Let unit vectors $v_1$ and $v_2$ be the $i^{th}$ principal vector pair for any two disjoint subspaces $S_1$ and $S_2$ in $\mathbb{R}^n$. Let the columns of the matrix $P \in \mathbb{R}^{n \times 2}$ be any two orthonormal vectors in the span of $v_1$ and $v_2$. Then for all vectors $x \in S_j$, $P^T x = \alpha t_j$ ($j \in \{1, 2\}$), where $\alpha \in \mathbb{R}$ depends on $x$ and $t_j \in \mathbb{R}^2$ is a fixed vector independent of $x$. Further, $\frac{t_1^T t_2}{\|t_1\|_2 \|t_2\|_2} = v_1^T v_2$*

*Proof: We use the notation $(M)_j$ to denote the $j^{th}$ column vector of matrix $M$ for any arbitrary matrix $M$. We claim that $t_j = P^T v_j$ ($j \in \{1, 2\}$). Also, without any loss of generality, assume that $(P)_1 = v_1$. Then in order to prove theorem 2, it suffices to show that $\forall x \in S_1$, $(P)_2^T x = 0$. By symmetry, $\forall x \in S_2$, $P^T x$ will also lie along a line in the subspace spanned by the columns of $P$.*

*Let the columns of $B_1 \in \mathbb{R}^{n \times d_1}$ and $B_2 \in \mathbb{R}^{n \times d_2}$ be the support of $S_1$ and $S_2$ respectively, where $d_1$ and $d_2$ are the dimensionality of the two subspaces. Then we can represent $v_1$ and $v_2$ as $v_1 = B_1 w_1$ and $v_2 = B_2 w_2$ for some $w_1 \in \mathbb{R}^{d_1}$ and $w_2 \in \mathbb{R}^{d_2}$. Let $B_1 w$ be any arbitrary vector in $S_1$ where*

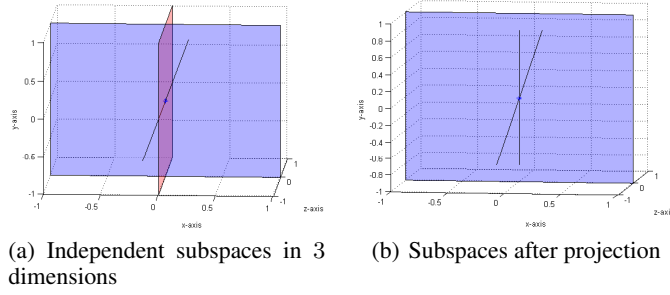

(a) Independent subspaces in 3 dimensions

(b) Subspaces after projection

Figure 1: **A three dimensional example of the application of theorem 2.** See text in section 3 for details.

$w \in \mathbb{R}^{d_1}$. Then we need to show that $T := (B_1 w)^T (P)_2 = 0 \forall w$. Notice that,

$$
\begin{aligned}
T &= (B_1 w)^T (B_2 w_2 - (w_1^T B_1^T B_2 w_2) B_1 w_1) \\
&= w^T (B_1^T B_2 w_2 - (w_1^T B_1^T B_2 w_2) w_1) \quad \forall w
\end{aligned}
\tag{2}
$$

*Let $USV^T$ be the svd of $B_1^T B_2$. Then $w_1$ and $w_2$ are the $i^{th}$ columns of $U$ and $V$ respectively, and $v_1^T v_2$ is the $i^{th}$ diagonal element of $S$ if $v_1$ and $v_2$ are the $i^{th}$ principal vectors of $S_1$ and $S_2$. Thus,*

$$
\begin{aligned}
T &= w^T (USV^T w_2 - S_{ii}(U)_i) \\
&= w^T (S_{ii}(U)_i - S_{ii}(U)_i) = 0 \quad \square
\end{aligned}
\tag{3}
$$

Geometrically, this theorem says that after projection on the plane $(P)$ defined by any one of the principal vector pairs between subspaces $S_1$ and $S_2$, both the entire subspaces collapse to just two lines such that points from $S_1$ lie along one line while points from $S_2$ lie along the second line. Further, the angle that separates these lines is equal to the angle between the $i^{th}$ principal vector pair between $S_1$ and $S_2$ if the span of the $i^{th}$ principal vector pair is used as $P$.

We apply theorem 2 on a three dimensional example as shown in figure 1. In figure 1 (a), the first subspace (y-z plane) is denoted by red color while the second subspace is the black line in x-y axis. Notice that for this setting, the x-y plane (denoted by blue color) is in the span of the $1^{st}$ (and only) principal vector pair between the two subspaces. After projection of both the entire subspaces onto the x-y plane, we get two lines (figure 1 (b)) as stated in the theorem.

Finally, we now show that for any $K$ class dataset with independent subspace structure, $2K$ projection vectors are sufficient for structure preservation.

**Theorem 3** *Let $X = \{x\}_{i=1}^N$ be a $K$ class dataset in $\mathbb{R}^n$ with Independent Subspace structure. Let $P = [P_1 \ldots P_K] \in \mathbb{R}^{n \times 2K}$ be a projection matrix for $X$ such that the columns of the matrix $P_k \in \mathbb{R}^{n \times 2}$ consists of orthonormal vectors in the span of any principal vector pair between subspaces $S_k$ and $\sum_{j \neq k} S_j$. Then the Independent Subspace structure of the dataset $X$ is preserved after projection on the $2K$ vectors in $P$.*

Before stating the proof of this theorem, we first state lemma 4 which we will use later in our proof. This lemma states that if two vectors are separated by a non-zero angle, then after augmenting these vectors with any arbitrary vectors, the new vectors remain separated by some non-zero angle as well. This straightforward idea will help us extend the two subspace case in theorem 2 to multiple subspaces.

**Lemma 4** *Let $x_1$, $y_1$ be any two fixed vectors of same dimensionality with respect to each other such that $\frac{x_1^T y_1}{\|x_1\|_2 \|y_1\|_2} = \gamma$ where $\gamma \in [0, 1)$. Let $x_2$, $y_2$ be any two arbitrary vectors of same dimensionality with respect to each other. Then there exists a constant $\bar{\gamma} \in [0, 1)$ such that vectors $x' = [x_1; x_2]$ and $y' = [y_1; y_2]$ are also separated such that $\frac{x'^T y'}{\|x'\|_2 \|y'\|_2} \leq \bar{\gamma}$.*

**Proof of theorem 3:**

---

**Algorithm 1** Computation of projection matrix $P$

---
**INPUT:** $X, K, \lambda, iter_{max}$

   **for** k=1 to K **do**

      $w_2^* \leftarrow$ random vector in $\mathbb{R}^{\bar{N}_k}$

      **while** $iter < iter_{max}$ or $\gamma$ not converged **do**

         $w_1^* \leftarrow \max_{w_1} \|X_k w_1 - \frac{\bar{X}_k w_2^*}{\|\bar{X}_k w_2^*\|_2}\|^2 + \lambda \|w_1\|^2$

         $w_1^* \leftarrow w_1^*/norm(w_1^*)$

         $w_2^* \leftarrow \max_{w_2} \|\frac{X_k w_1^*}{\|X_k w_1^*\|_2} - \bar{X}_k w_2\|^2 + \lambda \|w_2\|^2$

         $w_2^* \leftarrow w_2^*/norm(w_2^*)$

         $\gamma \leftarrow (X_k w_1^*)^T (\bar{X}_k w_2^*)$

      **end while**

      $P_k \leftarrow [X_k w_1^*, \bar{X}_k w_2^*]$

   **end for**

   $P^* \leftarrow [P_1 \ldots P_K]$

**OUTPUT:** $P^*$

---

*For the proof of theorem 3, it suffices to show that data vectors from subspaces $S_k$ and $\sum_{j \neq k} S_j$ (for any $k \in \{1 \ldots K\}$) are separated by margin less than 1 after projection using $P$. Let $x$ and $y$ be any vectors in $S_k$ and $\sum_{j \neq k} S_j$ respectively and the columns of the matrix $P_k$ be in the span of the $i^{th}$ (say) principal vector pair between these subspaces. Using theorem 2, the projected vectors $P_k^T x$ and $P_k^T y$ are separated by an angle equal to the the angle between the $i^{th}$ principal vector pair between $S_k$ and $\sum_{j \neq k} S_j$. Let the cosine of this angle be $\gamma$. Then, using lemma 4, the added dimensions in the vectors $P_k^T x$ and $P_k^T y$ to form the vectors $P^T x$ and $P^T y$ are also separated by some margin $\bar{\gamma} < 1$. As the same argument holds for vectors from all classes, the Independent Subspace Structure of the dataset remains preserved after projection.* $\square$

For any two disjoint subspaces, theorem 2 tells us that there is a two dimensional plane in which the entire projected subspaces form two lines. It can be argued that after adding arbitrary valued finite dimensions to the basis of this plane, the two projected subspaces will also remain disjoint (see proof of theorem 3). Theorem 3 simply applies this argument to each subspace and the sum of the remaining subspaces one at a time. Thus for $K$ subspaces, we get $2K$ projection vectors.

Finally, our approach projects data to $2K$ dimensions which could be a concern if the original feature dimension itself is less than $2K$. However, since we are only concerned with data that has underlying independent subspace assumption, notice that the feature dimension must be at least $K$. This is because each class must lie on at least 1 dimension which is linearly independent of others. However, this is too strict an assumption and it is straight forward to see that if we relax this assumption to 2 dimensions for each class, the feature dimensions are already at $2K$.

### 3.1 Implementation

A naive approach to finding projection vectors (say for a binary class case) would be to compute the SVD of the matrix $X_1^T X_2$, where the columns of $X_1$ and $X_2$ contain vectors from class 1 and class 2 respectively. For large datasets this would not only be computationally expensive but also be incapable of handling noise. Thus, even though theorem 3 guarantees the structure preservation of the dataset $X$ after projection using $P$ as specified, this does not solve the problem of dimensionality reduction. The reason is that given a labeled dataset sampled from a union of independent subspaces, we do not have any information about the basis or even the dimensionality of the underlying subspaces. Under these circumstances, constructing the projection matrix $P$ as specified in theorem 3 itself becomes a problem. To solve this problem, we propose an algorithm that tries to find the underlying principal vector pair between subspaces $S_k$ and $\sum_{j \neq k} S_j$ (for $k = 1$ to $K$) given the labeled dataset $X$. The assumption behind this attempt is that samples from each subspace (class) are not heavily corrupted and that the underlying subspaces are independent.

Notice that we are not specifically interested in a particular principal vector pair between any two subspaces for the computation of the projection matrix. This is because we have assumed independent subspaces and so each principal vector pair is separated by some margin $\gamma < 1$. Hence we

need an algorithm that computes any arbitrary principal vector pair, given data from two independent subspaces. These vectors can then be used to form one of the $K$ submatrices in $P$ as specified in theorem 3 . For computing the submatrix $P_k$, we need to find a principal vector pair between subspaces $S_k$ and $\sum_{j \neq k} S_j$. In terms of dataset $X$, we estimate the vector pair using data in $X_k$ and $\bar{X}_k$ where $\bar{X}_k := X \setminus \{X_k\}$. We repeat this process for each class to finally form the entire matrix $P^*$. Our approach is stated in algorithm 1. For each class $k$, the idea is to start with a random vector in the span of $\bar{X}_k$ and find the vector in $X_k$ closest to this vector. Then fix this vector and search of the closest vector in $\bar{X}_k$. Repeating this process till the convergence of the cosine between these 2 vectors leads to a principal vector pair. In order to estimate the closest vector from opposite subspace, we have used a quadratic program in 1 that minimizes the reconstruction error of the fixed vector (of one subspace) using vectors from the opposite subspace. The regularization in the optimization is to handle noise in data.

## 3.2 Justification

The definition 1 for margin $\gamma$ between two subspaces $S_1$ and $S_2$ can be equivalently expressed as

$$1 - \gamma = \min_{w_1, w_2} \frac{1}{2} \|B_1 w_1 - B_2 w_2\|^2 \ \ s.t. \ \|B_1 w_1\|^2 = 1, \|B_2 w_2\|^2 = 1 \tag{4}$$

where the columns of $B_1 \in \mathbb{R}^{n \times d_1}$ and $B_2 \in \mathbb{R}^{n \times d_2}$ are the basis of the subspaces $S_1$ and $S_2$ respectively such that $B_1^T B_1$ and $B_2^T B_2$ are both identity matrices.

**Proposition 5** *Let $B_1 \in \mathbb{R}^{n \times d_1}$ and $B_2 \in \mathbb{R}^{n \times d_2}$ be the basis of two disjoint subspaces $S_1$ and $S_2$. Then for any principal vector pair $(u_i, v_i)$ between the subspaces $S_1$ and $S_2$, the corresponding vector pair $(w_1 \in \mathbb{R}^{d_1}, w_2 \in \mathbb{R}^{d_2})$, s.t. $u_i = B_1 w_1$ and $v_i = B_2 w_2$, is a local minima to the objective in equation (4).*

*Proof: The Lagrangian function for the above objective is:*

$$\mathcal{L}(w_1, w_2, \boldsymbol{\eta}) = \frac{1}{2} w_1^T B_1^T B_1 w_1 + \frac{1}{2} w_2^T B_2^T B_2 w_2 - w_1^T B_1^T B_2 w_2 + \eta_1(\|B_1 w_1\|^2 - 1) + \eta_2(\|B_2 w_2\|^2 - 1) \tag{5}$$

*Then setting the gradient w.r.t. $w_1$ to zero we get*

$$\nabla_{w_1} \mathcal{L} = (1 + \eta_1) w_1 - B_1^T B_2 w_2 = 0 \tag{6}$$

*Let $USV^T$ be the SVD of $B_1^T B_2$ and $w_1$ and $w_2$ be the $i^{th}$ columns of $U$ and $V$ respectively. Then equation (6) becomes*

$$\begin{aligned} \nabla_w \mathcal{L} &= (1 + \eta_1) w_1 - USV^T w_2 \\ &= (1 + \eta_1) w_1 - S_{ii} w_1 = 0 \end{aligned} \tag{7}$$

*Thus the gradient w.r.t. $w_1$ is zero when $\eta_1 = 1 - S_{ii}$. Similarly, it can be shown that the gradient w.r.t. $w_2$ is zero when $\eta_2 = 1 - S_{ii}$. Thus the gradient of the Lagrangian $L$ is $0$ w.r.t. both $w_1$ and $w_2$ for every corresponding principal vector pair. Thus vector pair $(w_1, w_2)$ corresponding to any of the principal vector pairs between subspaces $S_1$ and $S_2$ is a local minima to the objective 4.* $\square$

Since $(w_1, w_2)$ corresponding to any principal vector pair between two disjoint subspaces form a local minima to the objective given by equation (4), one can alternatively minimize equation (4) w.r.t. $w_1$ and $w_2$ and reach one of the local minima. Thus, by assuming independent subspace structure for all the $K$ classes in algorithm 1 and setting $\lambda$ to zero, it is straight forward to see that the algorithm yields a projection matrix that satisfies the criteria specified by theorem 3.

Finally, real world data do not strictly satisfy the independent subspace assumption in general and even a slight corruption in data may easily lead to the violation of this independence. In order to tackle this problem, we add a regularization ($\lambda > 0$) term while solving for the principal vector pair in algorithm 1. If we assume that the corruption is not heavy, reconstructing a sample using vectors belonging to another subspace would require a large coefficient over those vectors. The regularization avoids reconstructing data from one class using vectors from another class that are slightly corrupted by assigning such vectors small coefficients.

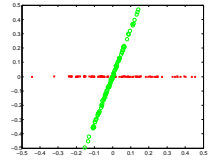

(a) Data projected using $P_a$

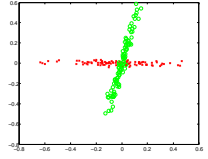

(b) Data projected using $P_b$

Figure 2: **Qualitative comparison between (a)** *true* **projection matrix and (b) projection matrix from the proposed approach on high dimensional synthetic two class data.** See section 4.1.1 for details.

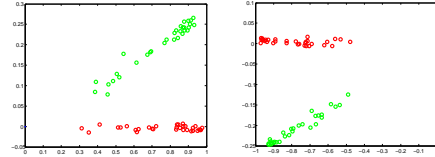

(a)        (b)

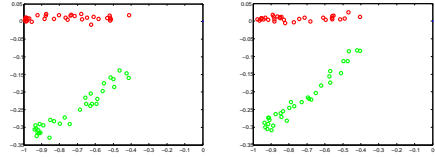

(c)        (d)

Figure 3: **Four different pairs of classes from the Extended Yale dataset B projected onto two dimensional subspaces using proposed approach.** See section 4.1.1 for details.

### 3.3 Complexity

Solving algorithm 1 requires solving an unconstrained quadratic program within a while-loop. Assume that we run this while loop for T iterations and that we use conjugate gradient descent to solve the quadratic program in each iteration. Also, it is known that for any matrix $A \in \mathbb{R}^{a \times b}$ and vector $b \in \mathbb{R}^a$, conjugate gradient applied to a problem of the form

$$\arg \min_w \|Ax - b\|^2 \tag{8}$$

takes time $\mathcal{O}(ab\sqrt{\mathcal{K}})$, where $\mathcal{K}$ is the condition number of $A^T A$. Thus it is straight forward to see that the time required to compute the projection matrix for a $K$ class problem in our case is $\mathcal{O}(KTnN\sqrt{\mathcal{K}})$, where $n$ is the dimensionality of feature space, $N$ is the total number of samples and $\mathcal{K}$ is the condition number of the matrix $(X_k^T X_k + \lambda \mathcal{I})$. Here $\mathcal{I}$ is the identity matrix.

## 4 Empirical Analysis

In this section, we present empirical evidence to support our theoretical analysis of our subspace learning approach. For real world data, we use the following datasets:

*1. Extended Yale dataset B [3]*: It consists of $\sim 2414$ frontal face images of 38 individuals ($K = 38$) with 64 images per person. These images were taken under constrained but varying illumination conditions.

*2. AR dataset [10]*: This dataset consists of more than 4000 frontal face images of 126 individuals with 26 images per person. These images were taken under varying illumination, expression and facial disguise. For our experiments, similar to [15], we use images from 100 individuals ($K = 100$) with 50 males and 50 females. We further use only 14 images per class which correspond to illumination and expression changes. This corresponds to 7 images from Session 1 and rest 7 from Session 2.

*3. PIE dataset [12]*: The pose, illumination, and expression (PIE) database is a subset of CMU PIE dataset consisting of $11,554$ images of 68 people ($K = 68$).

We crop all the images to $32 \times 32$, and concatenate all the pixel intensity to form our feature vectors. Further, we normalize all data vectors to have unit $\ell^2$ norm.

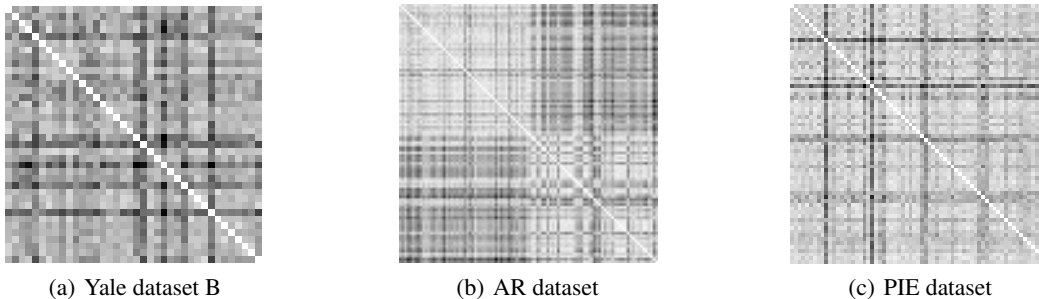

|             |             |             |
|:-----------:|:-----------:|:-----------:|
| (a) Yale dataset B | (b) AR dataset | (c) PIE dataset |

Figure 4: **Multi-class separation after projection using proposed approach for different datasets.** See section 4.1.2 for details.

## 4.1 Qualitative Analysis
### 4.1.1 Two Subspaces-Two Lines

We test both the claim of theorem 2 and the quality of approximation achieved by algorithm 1 in this section. We perform these tests on both synthetic and real data.

*1. Synthetic Data:* We generate two random subspaces in $\mathbb{R}^{1000}$ of dimensionality 20 and 30 (notice that these subspaces will be independent with probability 1). We randomly generate 100 data vectors from each subspace and normalize them to have unit length. We then compute the $1^{st}$ principal vector pair between the two subspaces using their basis vectors by performing SVD of $B_1^T B_2$, where $B_1$ and $B_2$ are the basis of the two subspaces. We orthonormalize the vector pair to form the projection matrix $P_a$. Next, we use the labeled dataset of 200 points generated to form the projection matrix $P_b$ by applying algorithm 1. The entire dataset of 200 points is then projected onto $P_a$ and $P_b$ separately and plotted in figure 2. The green and red points denote data from either subspace. The results not only substantiate our claim in theorem 2 but also suggest that the proposed algorithm for estimating the projection matrix is a good approximation.

*2. Real Data:* Here we use Extended Yale dataset B for analysis. Since we are interested in projection of two class data in this experimental setup, we randomly choose 4 different pairs of classes from the dataset and use the labeled data from each pair to generate the two dimensional projection matrix (for that pair) using algorithm 1. The resulting projected data from the 4 pairs can be seen in figure 3. As is evident from the figure, the projected two class data for each pair approximately lie along two different lines.

### 4.1.2 Multi-class separability

We analyze the separation between the $K$ classes of a given $K$-class dataset after dimensionality reduction. First we compute the projection matrix for that dataset using our approach and project the data. Second, we compute the top principal vector for each class separately from the projected data. This gives us $K$ vectors. Let the columns of the matrix $Z \in \mathbb{R}^{2K \times K}$ contain these vectors. Then in order to visualize inter-class separability, we simply take the dot product of the matrix $Z$ with itself, i.e. $Z^T Z$. Figure 4 shows this visualization for the three face datasets. The diagonal elements represent self-dot product; thus the value is 1 (white). The off-diagonal elements represent inter-class dot product and these values are consistently small (dark) for all the three datasets reflecting between class separability.

## 4.2 Quantitative Analysis

In order to evaluate theorem 3, we perform a classification experiment on all the three real world datasets mentioned above after projecting the data vectors using different dimensionality reduction techniques. We compare our quantitative results against PCA, Linear discriminant analysis (LDA), Regularized LDA and Random Projections (RP) [1]. We make use of sparse coding [15] for classification.

For Extended Yale dataset B, we use all 38 classes for evaluation with $50\% - 50\%$ train-test split 1 and $70\% - 30\%$ train-test split 2. Since our method is randomized, we perform 50 runs of computing the projection matrix using algorithm 1 and report the mean accuracy with standard deviation. Similarly for RP, we generate 50 different random matrices and then perform classification. Since all other methods are deterministic, there is no need for multiple runs.

Table 1: **Classification Accuracy on Extended Yale dataset B with 50%-50% train-test split.** See section 4.2 for details.

| Method | Ours | PCA | LDA | Reg-LDA | RP |
|---|---|---|---|---|---|
| dim | 76 | 76 | 37 | 37 | 76 |
| acc | $\mathbf{98.06 \pm 0.18}$ | 92.54 | 83.68 | 95.77 | $93.78 \pm 0.48$ |

Table 2: **Classification Accuracy on Extended Yale dataset B with 70%-30% train-test split.** See section 4.2 for details.

| Method | Ours | PCA | LDA | Reg-LDA | RP |
|---|---|---|---|---|---|
| dim | 76 | 76 | 37 | 37 | 76 |
| acc | $\mathbf{99.45 \pm 0.20}$ | 93.98 | 93.85 | 97.47 | $94.72 \pm 0.66$ |

Table 3: **Classification Accuracy on AR dataset.** See section 4.2 for details.

| Method | Ours | PCA | LDA | Reg-LDA | RP |
|---|---|---|---|---|---|
| dim | 200 | 200 | 99 | 99 | 200 |
| acc | $\mathbf{92.18 \pm 0.08}$ | 85.00 | - | 88.71 | $84.76 \pm 1.36$ |

Table 4: **Classification Accuracy on a subset of CMU PIE dataset.** See section 4.2 for details.

| Method | Ours | PCA | LDA | Reg-LDA | RP |
|---|---|---|---|---|---|
| dim | 136 | 136 | 67 | 67 | 136 |
| acc | $\mathbf{93.65 \pm 0.08}$ | 87.76 | 86.71 | 92.59 | $90.46 \pm 0.93$ |

Table 5: **Classification Accuracy on a subset of CMU PIE dataset.** See section 4.2 for details.

| Method | Ours | PCA | LDA | Reg-LDA | RP |
|---|---|---|---|---|---|
| dim | 20 | 20 | 9 | 9 | 20 |
| acc | $\mathbf{99.07 \pm 0.09}$ | 97.06 | 95.88 | 97.25 | $95.03 \pm 0.41$ |

For AR dataset, we take the 7 images from Session 1 for training and the 7 images from Session 2 for testing. The results are shown in table 3. The result using LDA is not reported because we found that the summed within class covariance was degenerate and hence LDA was not applicable. It can be clearly seen that our approach significantly outperforms other dimensionality reduction methods.

Finally for PIE dataset, we perform experiments on two different subsets. First, we take all the 68 classes and for each class, we randomly choose 25 images for training and 25 for testing. The performance for this subset is shown in table 4. Second, we take only the first 10 classes of the dataset and of all the 170 images per class, we randomly split the data into $70\% - 30\%$ train-test set. The performance for this subset is shown in table 5.

Evidently, our approach consistently yields the best performance on all the three datasets compared to other dimensionality reduction methods.

# 5   Conclusion

We proposed a theoretical analysis on the preservation of independence between multiple subspaces. We show that for $K$ independent subspaces, $2K$ projection vectors are sufficient for independence preservation (theorem 3). This result is motivated from our observation that for any two disjoint subspaces of arbitrary dimensionality, there exists a two dimensional plane such that after projection, the entire subspaces collapse to just two lines (theorem 2). Resulting from this analysis, we proposed an efficient iterative algorithm (1) that tries to exploit these properties for learning a projection matrix for dimensionality reduction that preserves independence between multiple subspaces. Our empirical results on three real world datasets yield *state-of-the-art* results compared to popular dimensionality reduction methods.

## Footnotes

[1]We also used LPP (Locality Preserving Projections) [4], NPE (Neighborhood Preserving Embedding) [5], and Laplacian Eigenmaps [1] for dimensionality reduction on Extended Yale B dataset. However, because the best performing of these reduction techniques yielded a result of only 73% compared to the close to 98% accuracy from our approach, we do not report results from these methods.

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
