[Reviews · NeurIPS 2014]

Submitted by Assigned_Reviewer_11

The authors present an efficient method for dimensionality reduction which preserves subspace independence and, as consequence, is useful for problems such as classification where we assume that the data are separable on a lower-dimensional subspace. They show that the subspace structure preservation can be achieved with 2K projects, and they perform a detailed comparison with other methods on benchmark data.

The paper is very thorough and well-presented. It presents a novel and effective method for an important problem in dimensionality reduction, and it does so in a way that makes it clear that the method is competitive with popular alternatives. As such, I think it is an excellent paper.

Some small comments:

Lemma 4 seems to have remained without proof. I don't think it is incorrect, but including the proof (even if short) would strengthen the paper further.
I hadn't come across the sigma summation operator for subspaces, and its meaning was not immediately apparent; maybe the authors should define this operation to aid reader comprehension.
On page 7 (or alternatively in the Conclusion) I would have liked the authors to say something about the methodological differences between their method and the methods being compared.
In the tables, for the randomized methods it would be good to include the standard error (although admittedly the scores are high enough for this not to be crucial).

Summary: An excellent paper that advances the field in significant ways.

Submitted by Assigned_Reviewer_42

This paper presents a method for dimensionality reduction in situations where it is assumed that the data are sampled from a union of independent subspaces. The dimensionality reduction algorithm preserves this subspace relationship in the lower dimensional space. For the general case when the subspaces are unknown, a method for estimating the projection matrix is proposed. This method is derived based on a key idea of this work that data from K classes can be transformed using an independence preserving projection matrix composed of 2K vectors.

Quality:
This paper shows a nice way of doing dimensionality reduction while preserving subspace properties of data. In some ways, it is similar to the neighborhood-preserving dimensionality reduction techniques except working with a subspace representation. The projection is learned via an iterative method when the underlying subspaces of the data are unknown (typical situation).

The empirical results seem quite strong, though it would have been better to evaluate on a variety of tasks rather than 3 different face recognition tasks. Also, reporting the standard deviation of the 50 trials would also be helpful.

It seems like one limitation of this method is that it is only applicable when the dimension of the input data is greater than twice the number of classes, since the projection size is fixed at 2K x n. This seems like a significant limitation for this to be a general purpose dimensionality reduction tool. Also, there is a claim that the regularization in the learning of the projection matrix P helps the algorithm be robust to corrupted data. However, it would be good if this claim was experimentally validated.

Summary: This method proposes an useful method for dimensionality reduction which preserves independence between constituent subspaces. Strong results compared to other dimensionality reduction techniques are reported but the benefits of this approach are limited to cases where the dimensionality of the data is less than the twice the number of classes.

Submitted by Assigned_Reviewer_45

The paper presents a dimensionality reduction method based for data that lives in a union of independent subspaces. The method is based on the observation that two independent subspaces collapse into two lines if a projection consisting of one vector from each subspace is applied on the data. The authors then present an algorithm to find vectors from different subspaces to be used for projection. Experimental results show that the proposed method performs better than state-of-the-art.

In general, I believe that the paper presents a new approach, it explores a new idea, and it provides very good experimental results, and therefore eventually it's worth publishing. However, in my opinion, the paper is not written very well. The authors stress in multiple places that the idea that 2K projection vectors are sufficient for K class data is "non-trivial". This might be the case, but it's a very intuitive result. I'm very surprised that the paper first presents a formal proof before the geometrical explanation, and this geometrical explanation is not very clear, and Figure 1 is very poorly designed. The proof is also very convoluted, for instance the same subscript j is used to both denote the components of vectors & matrices, as well as different subspaces. Section 3 can be re-organized into a theory and algorithm section, the current structure is not making this distinction clear.

The authors can give simple directions how to go about proving Lemma 4 (why is this not Lemma 1?) In the same lemma, there's no guarantee that gamma_hat != 0. Is this a problem? Please comment. In line 179, I wouldn't use \Sigma for subspaces. Or if you do, it'll be good to briefly explain what addition of subspaces mean.

It is clear that two major potential problem for the proposed method is noise, and cases where the independent subspaces assumption is not valid. I think the paper doesn't provide enough empirical evidence to investigate these cases. The study in Section 4.1 just verifies the theorems presented in the previous sections, which are already proven. What happens when the independence subspace assumption is not valid?

It is not clear what value Section 4.1.2 and Figure 4 have. These can potentially be removed and a more-in-depth analysis can be added instead.

The algorithms compared with in Section 4.2 are not state-of-the-art, not even close. There has been a large progress in this area (low rank and sparse subspace clustering), and I think it's unacceptable to ignore all of these recent methods and just compare with older methods like PCA and LDA..
Summary: While the paper presents an interesting result and algorithm, the paper is not very well written and empirical results do not contain comparisons with the state-of-the-art in this area.
Author Feedback
Author rebuttal: We would like to thank all the reviewers for their productive feedback.

Assigned Reviewer (AR) 11 and 45:
1. We omitted the proof of lemma 4 due to spatial constraints. We will try to accommodate it in the final version (if accepted).
2. Summation of subspaces denotes the set of linear combination of vectors, each of which belongs to the subspaces used in the summation.

AR 11 and 42:
1. We will include the standard deviations of errors in the tables in the final version.

AR 42:
1. Reviewer expresses his/her concern with the case when the dimensionality of data is less than twice the number of classes (to which our approach projects the original data). Although this might seem like a genuine concern at first, we will clarify why this is not a problem in most real world problems. Since the paper is only concerned with data that has underlying independent subspace assumption, notice that the original dimension of the data must be at least K (#classes). This is because each class must lie on at least 1 dimension which is linearly independent of others. However, this is too strict an assumption and it is straight forward to see that if we relax this assumption to 2 dimensions for each class, the original dimensions are already at 2K. Thus original data dimensions being less than 2K should therefore not practically be a concern.

AR 42 and 45:
1. Our claim that the proposed method is capable of handling noise in data has indeed been empirically corroborated by the face datasets' classification experiments. Notice that while faces are generally assumed to lie along low dimensional subspaces, they practically tend to be very noisy and do not strictly follow this assumption. To add to this point, figure 3,4/section 4.1 (visualization of 2 class/multi-class projected data using our technique) also shows our method's robustness towards noise. So we have provided both qualitative and quantitative evidence.

AR 45:
1. \gamma_hat=0 is actually the best case scenario (notice that \gamma_hat=0 implies that subspaces are maximally separated, kindly refer the Preliminaries section). So no, \gamma_hat=0 is not a problem.
2. As mentioned above, section 4.1 (figures 3,4) doesn't simply verify theorem 2 (no noise assumption) but also shows the capability of our approach to handle noise, as face datasets are known to be heavily corrupted (see 'Robust Subspace Segmentation by Low-Rank Representation' for example) given the strict independent subspace assumption.
3. In this paper, it is very important to note that we are interested in the analysis of data with independent subspace assumption with/without noise ONLY. This is a key assumption we make in the paper, therefore, we are not concerned with cases where this assumption clearly does not hold. This is why we perform our evaluation on different face datasets.
4. As a footnote on page 7, we did mention that we compared our approach with Locality Preserving Projections, Neighborhood Preserving Embedding and Laplacian Eigenmaps. However since these techniques were not competent (the maximum accuracy of the three algorithms was 73%) compared to the other techniques we compared with (~95% accuracy), we decided to drop them in our overall evaluation. This comparatively poor result is not surprising because the aforementioned dimensionality reduction techniques do not guarantee the preservation of subspace structure; hence they fail to perform well under this assumption.
5. We were confused with reviewer's statement about comparing our approach with 'low rank and sparse subspace clustering'. Those are clustering techniques while we have presented a dimensionality reduction technique, though related are clearly not the same.

AR 11, 42 and 45:
We would like to mention that a dimensionality reduction technique specifically aimed at preserving independent subspace structure in data has not been studied so far in the literature, to the best of our knowledge. In this paper, we have established a relation between the number of classes and the sufficient number of vectors required for dimensionality reduction, and we believe that this analysis holds importance for further research in this area. Practically, as the size of datasets (specifically data with the independent subspace assumption) continues to increase, an optimized version of an algorithm such as we propose becomes extremely useful in analyzing such data, for example, large-scale face recognition.